# Human Colorectal Carcinoma Infiltrating B Lymphocytes Are Active Secretors of the Immunoglobulin Isotypes A, G, and M

**DOI:** 10.3390/cancers11060776

**Published:** 2019-06-04

**Authors:** Christina Susanne Mullins, Michael Gock, Mathias Krohn, Michael Linnebacher

**Affiliations:** 1Department of General Surgery, Molecular Oncology and Immunotherapy, University Medicine Rostock, Schillingallee 69, 18057 Rostock, Germany; christina.mullins@med.uni-rostock.de (C.S.M.); mathias.krohn@med.uni-rostock.de (M.K.); 2Department of General Surgery, University Medicine Rostock, Schillingallee 35, 18057 Rostock, Germany; michael.gock@med.uni-rostock.de

**Keywords:** B cells, tumor infiltrating lymphocytes, tumor microenvironment, FluoroSpot, colorectal carcinoma

## Abstract

Despite the importance of tumor infiltrating B cells (TiBc) in immunological circuits, their functional role is scarcely investigated. Here, we analyzed immunoglobulin (Ig) secretion of the subtypes IgA, IgG, and IgM of TiBc from freshly resected primary and secondary colorectal carcinomas (CRC) by FluoroSpot (*n* = 30 CRC) directly ex vivo. High, intermediate, and low secretion was observed in 33%, 37%, and 30% of the tumors for IgA; in 10%, 27%, and 63% for IgG; and in 21%, 36%, and 50% for IgM, respectively. These ex vivo data validate our previous findings: Most TiBc present in the CRC microenvironment are functional since they produce and actively secrete Ig (IgA > IgG > IgM). Of note, the presence of major histocompatibility complex (MHC) class II expressing cells in the tumor micromilieu only correlated with IgG secretion (*p* = 0.0004). Supporting recent findings in several other tumor entities, TiBc in CRC thus likely can contribute to tumor control in a dual role of sole antigen-presentation and additionally anti-tumoral Ig-production.

## 1. Introduction

The concept that the immune system is involved in the combat against cancer is not new; however, this notion has only recently spread widely in the cancer biology community, especially due to the groundbreaking revelation of cancer immunoediting mechanisms [1]. The delineation of the three stages—elimination, equilibrium, and escape [1]—have set the stage for immune checkpoint therapies. A precise examination of the spatio-temporal dynamics of immune cell types infiltrating tumors unraveled the infiltrates’ composition and thus was dubbed the immunome [2]. For colorectal carcinomas (CRC), two immunome clusters could be identified. The critical players associated with prolonged disease-free survival belonged to a T cell subset of Th1, Tγδ, and cytotoxic T cells in addition to macrophages and mast cells. Furthermore, the cluster included cells expressing major histocompatibility complex (MHC) class II and B cell co-stimulatory markers [2]. Most of the T cell subpopulations were found in early stage CRC and T cell numbers decreased with tumor progression, whereas the density of B cells increased with the tumor stage [2]. Until recently, B cells have not been, for the most part, considered an important population of tumor-infiltrating lymphocytes (TIL). Yet, up to 40% of the TIL may be B cells [3,4]. Recent findings suggest that tumor-infiltrating B cells (TiBc) can elicit a substantial humoral anti-tumor response, facilitate T cell responses, serve as antigen-presenting cells, or even possess direct anti-tumoral cytotoxicity [4]. Of note, B cells possess the unique capability of being able to concentrate antigens through membrane immunoglobulin (Ig) mediated uptake, which might also facilitate T cell activation above certain thresholds for tumor-specific antigens [4,5]. Furthermore, they are involved in the formation of tertiary lymphoid structures that sustain long-term immunity [6].

Recently, we demonstrated that Epstein Barr virus (EBV)-immortalized TiBc cultures have an activated immune phenotype (CD23+, CD80+) and that at least some clones produce immunoglobulins (Ig) capable of specifically recognizing and binding to CRC cells [7]. However, we could not formally rule out the possibility that these features were influenced by the EBV-transformation process. Thus, the major goal of the present study was to analyze the Ig-secretion behavior of “untouched” TiBc fresh from CRC, which incontrovertibly represents the in situ situation with minimum bias due to cell handling and cultivation.

## 2. Results

### 2.1. Patient Characteristics

In total, 25 CRC patients were included in this study. Two patients suffered from rectal cancer, 18 from colon cancer (two of them with synchronous metastases); one patient had a multifocal CRC including the colon and rectum (HROC252). Out of four patients with metachronous metastases, two had a rectal and two a colonic cancer in history. All of these patients received several courses of chemotherapy prior to metastasectomy. There were 9 female and 16 male patients with a median age of 69 years (ranging from 30 to 92 years, Table 1). The UICC stage varied between stage IIa and IV. Radical resection of the primary tumor was achieved in every patient. In analyzing the primary tumor site, 8 right sided (coecum, ascendens, right flexure) and 12 left sided (left flexure, descendens, sigmoid, recto-sigmoid) colonic cancers were resected. Two resected rectal tumors were located in the upper and one in the middle third of the rectum (Table 1). The adenoma case, HROC263, was not included in this listing.

The mean time of follow up for all patients was 3.5 ± 1.2 years and the overall survival rate was 48%, while all patients suffering from stage UICC IV CRC died after 3.9 ± 1.4 years (Table 1).

### 2.2. Detailed Analyses of Tumor Tissue Composition and Ig Secretion in CRC Patients

The cellular composition of the in situ analyzed colorectal specimen (*n* = 27/30) was assessed by flow cytometry using the epithelial cell markers: CD326, which reliably stains almost all CRC cells; MHC class II as marker of antigen presenting cells; CD3 for T cells; and a lineage panel consisting of CD19, CD20, and CD21 to detect B cells. CD326, which served as a CRC marker in this study, was detected in nearly all cell suspensions of the fresh tumor tissue (Figure 1A). Only three cell suspensions lacked a substantial presence of CD326 positive cells (HROC111Met, HROC230, and HROC248, Figure 1A). Roughly a sixth of cells in the suspension were antigen presenting cells (APC) represented by MHC II+ and about 2% to 3% of the cells were B (CD19, CD20, and/or CD21+) and T (CD3+) cells, respectively (Figure 1A). The percentage of B cells ranged from 0% to 8%. In two samples (HROC251 and HROC278), no B cells were detectable by flow cytometry.

Subsequently, freshly prepared single cell suspensions of all included specimens were analyzed for the secretion of IgA, IgG, and IgM. These immunoglobulin subclasses were previously found to be produced by cultured TiBc clones (Figure 1B). In total, 29 primary and metastatic tumor tissues and one adenoma (HROC263) could be analyzed. Out of the 30 tissues analyzed, 10 exhibited a high, 11 an intermediate, and 9 low to no IgA secretion. For IgG, 3 tissues were found to have high levels, 8 intermediate levels, and 19 only very low amounts of Ig secretors. In the case of IgM, we identified 6/28 samples with high, 10/28 with intermediate, and 14/28 with low/no secretors. In total, 22/30 samples contained cells secreting at least one type of Ig, 5 samples contained cells secreting two different isotypes, and for 6 samples we observed substantial secretion of all three Ig isotypes.

Three synchronal tumors of a 45 year old male Lynch patient who underwent subtotal colectomy were analyzed for Ig secretion. All tumors were microsatellite instable (MSI) adenocarcinomas, and the highest tumor stage according to the UICC classification was IIb (G3 pT4b pN0 (0/74) L0 V1 R0 cM0). Tumor 1 was localized in the descending colon, tumor 2 in the sigmoid colon, and tumor 3 in the rectum. For all three tumors, we observed substantial secretion of all three Ig isotypes (Figure 1C).

### 2.3. Analyses of Primary CRC Tumors and CRC Metastasis

Comparison of the expression and secretion profiles of primary CRC tumor to CRC metastasis revealed a tendency towards higher overall Ig secretion for primary tumors (Figure 2A); however, this failed to reach statistical significance. The composure of the tissues (expression profile assessed by flow cytometry) did not differ (Figure 2B).

The direct comparison of Ig secretion from the tissues of two patients with primary CRC and corresponding synchronous liver metastases (HROC277 and HROC278) also revealed no differences between primary or metastatic CRC. However, in both cases, neither the primary nor the corresponding metastases secreted Ig of any class.

### 2.4. Correlation of Tumor Tissue Composition (Including the Immune Cell Populations) with Ig Secretion

Potential correlations between tumor tissue composition (as assessed by flow cytometry; including the most relevant immune cell populations) and Ig secretion (as determined by FluoroSpot analyses) were assessed. Here, we found that MHC II expression is highly significantly correlated (*p* = 0.0004) with IgG secretion (Figure 3A). A trend (*p* = 0.0647) towards increased IgG secretion in combination with a higher expression of B cell lineage markers (CD19, 20, and 21) was observed (Figure 3B). In contrast, neither IgA nor IgM expression correlated with any of the tumor tissue characteristics, including markers for (infiltrating) immune cells.

## 3. Discussion

The recognition and study of B cells as part of the TIL compartment is a rather recent event (and accelerated at the beginning of this decade). Despite studies demonstrating a tumor-protective or -promoting role of B cells [8], TiBc undeniably can also contribute to the combat against cancer [9,10]. They show an activated immune phenotype (CD23+, CD27+, CD80+ [10,11]) and are capable of tumor-specific Ig production [7]. Tumor-associated B cells were enriched for activated and terminally differentiated B cells. Very recently, Spear and colleagues shed light onto the allegedly divergent pro- or anti-tumoral role of TiBc in a matched analysis of a genetic and an orthotopic cell injection model of murine pancreatic carcinoma [12]. They concluded that TiBc do not support tumor growth and are an active member of the tumor microenvironment in the genetic mouse model, which best reflects (slow) natural human tumor development. The present study is the first to address the functionality of TiBc fresh from human tumors. We demonstrate not only that TiBc fresh from resection specimen (= in situ) secrete Ig of the isotypes A, G, and M, but also show that the secretion of IgG (but not IgA or IgM) is positively correlated with antigen-presenting cell (APC) infiltration (represented by MHC II expression; *p* = 0.0004). IgG secretion additionally tends to be linked to the degree of TiBc infiltration in the resection specimen (*p* = 0.0647). Of note, the infiltration of CRC with B lymphocytes was further correlated with that of T lymphocytes (*p* = 0.0282) in our CRC series. A relevant proportion of regulatory B cells can only be detected in advanced CRC tumors and metastases [11]. Thus, B cells are very capable of aiding in the immunological combat against cancer and thus the group of Pagès and Gallon stated, already in 2010, that the immune infiltration in human tumors serves as a prognostic factor that should not be ignored [13].

In a limited sample size of synchronous tumors and/or metastases (*n* = 3 patients; HROC252, HROC277, and HROC278), we found high similarities with regard to the corresponding tumor tissue composition and Ig secretion. When comparing the secretion of primary CRC tumors to that of the matching metastases, in general, primaries tend to have a higher overall Ig secretion (29.2 vs. 5.5, 3.1 vs. 0.5, and 13.8 vs. 3.5 average specific spots for IgA, IgM, and IgG, respectively). While this observation fails to reach statistical significance, it is in line with the finding that the percentage of B cells within tumors is higher than that in the peripheral blood of CRC patients, and metastases are typically devoid of tumor-infiltrating B cells [11]. The composure of the primary and metastatic tumoral tissues does not differ—with the exception of a tendency towards a higher APC infiltration of primary CRC (19.4% vs. 5.2% MHC II positive cells); however, this again fails to reach statistical significance. In total, our data support the findings of previous reports on the pre-activated status of TiBc [4,6,7]. The fact that in CRC specimen generally more IgG than IgM is secreted in situ further corroborates previous antigen-specific maturation of the majority of TiBc.

Due to the fact that we prepared (single) cell suspensions of the tissues for expression and secretion analyses, we cannot attribute certain properties to a precise cell population in situ. However, the tumor pieces used for analyses were cut by a trained pathologist exclusively out of the invasive front. Additionally, a high tumor cell content was a mandatory prerequisite for subsequent detailed analysis. Thus, the physical origin of B cells can be narrowed down to either “truly” tumor-infiltrating or to, at least, near to the infiltrative border. Of course, we assume that a proportion of the lymphocytes may also originate from vessels within the tumor tissue, which might explain the relative high proportion of IgM-secreting TiBc in some of the cases. Admittedly, the IgM-secretion is slightly contradictory to previous studies, which described TiBc as IgM-negative [10,14]. Yet, the quantity of erythrocytes in the analyzed tumor preparations was generally low and thus indicates only limited contaminating blood from tumor vascularization. One question we did not address is how the proportion of Ig secreting cells in tumor tissues compares to that in the normal gut of the same patients. In normal gut tissue, IgA producers are, with 90%, by far the most abundant B cell type [15]. Whereas IgA-producing TiBc counted as only around 63% in the CRC cell suspensions, the proportion of B cells secreting IgM was in the range of normal tissue (7% vs. 6%). Of note, numbers of IgG-producing TiBc were much higher than IgG-producing B cells in healthy human gut (on average approximately 30% vs. 4% [15]). When comparing the results of the present ex vivo/in situ analysis with the Ig secretion observed for EBV-immortalized TiBc clones in our previous study [7], no striking differences became evident concerning Ig subclass distribution. Thus, EBV transformation of TiBc to generate easy expandable clones is likely an interesting technique since it allows for a broad technical spectrum of TiBc investigations, including the functional deduction of TiBc-derived Ig and the identification of (tumor-specific) target epitopes.

## 4. Materials and Methods

### 4.1. Patients

Patients treated at the university hospital of Rostock in the years 2012 to 2014 were included in this study. All CRC resection specimens were received fresh from surgery with informed written patient consent. In total, 23 patients suffered from CRC and underwent surgical resection of the tumor after initial diagnosis. In two of these patients, tumor metastases (liver, peritoneum) were resected during the primary operation. One adenoma case was additionally included. Three patients suffered from metachronous liver metastases and one patient suffered from CRC brain metastases. These metastases were resected after neoadjuvant chemotherapy. All procedures were approved by the Ethics Committee of the University of Rostock (reference number II HV 43/2004) in accordance with generally accepted guidelines for the use of human material.

### 4.2. Tumor Preparation

Tumor tissue was received fresh from surgery and a single cell suspension was prepared immediately, as described previously [16]. Briefly, the tissue was minced by crossed scalpels and passed through a cell strainer to obtain a single cell suspension.

### 4.3. Flow Cytometry

In total, 2 × 10^5^ suspension cells were stained with 4 µg of the respective antibody (mix 1: CD3 APC, CD4 FITC, and CD8 PE; mix 2: CD90 FITC, MHC I PE, and MHC II APC; mix 3: CD19, CD20, and CD21 FITC, CD326 PE). Antibody incubation was performed for 30 minutes on ice, followed by a wash step with PBS. Stained cell suspensions were analyzed immediately using a FACSCalibur (BD, Heidelberg, Germany). Irrelevant antibodies of the same isotype served as controls. All antibodies used were obtained from Immunotools (Friesoythe, Germany) with the exception of CD326, which was obtained from Miltenyi Biotec GmbH (Bergisch Gladbach, Germany).

### 4.4. FluoroSpot

The principal of a FluoroSpot assay is the same as that of an ELISpot, but the detection antibodies are labeled with fluorochromes, resulting in a higher sensitivity and lower background. For our FluoroSpot analyses, 1 × 10^5^ suspension cells per well were seeded in a 96 well FluoroSpot plate pre-coated with anti IgA, anti IgG, and anti IgM antibodies (all reagents and protocols from Mabtech, Nacka Strand, Sweden) and incubated at 37 °C and 5% CO_2_ for 72 to 144 h. Plates were analyzed and an automated fluorescent spot counting was done on a FluoroSpot Analyzer (Cellular Technology Limited (CTL), Cleveland, OH, USA). Analyses were performed in duplicates. The mean number of spots per sample and Ig type were plotted in a heat map using the conditional formatting function of Excel (Microsoft Excel 2010, Microsoft Corporation, Redmond, WA, USA). Low numbers of secreting cells per Ig type are represented by red shading, intermediate numbers are represented by very light shading, and high numbers by blue shading. Proper functionality of the FluoroSpot was ensured by using additional wells containing TiBc clones known to secrete Ig of the respective class [7] or containing only medium (negative control).

### 4.5. Statistical Analyses

All statistical analyses (*t*-test for comparison of primary and metastatic samples and Pearson r for correlation of Ig secretion (IgA, IgG, and IgM) and tumor tissue composition (T cells, B cells, and MHC II^+^ cells) were performed using Prism5 (GraphPad, San Diego, CA, USA). Dot plot (including SEM) graphs represent the number of Ig-producing cells of primary and metastatic samples. *p*-values lower than 0.05 were considered significant.

## 5. Conclusions

TiBc most likely contribute to tumor control in a dual mechanism of (I) antigen-presentation and (II) anti-tumoral Ig-production in several tumor entities, including CRC.

## Figures and Tables

**Figure 1 cancers-11-00776-f001:**
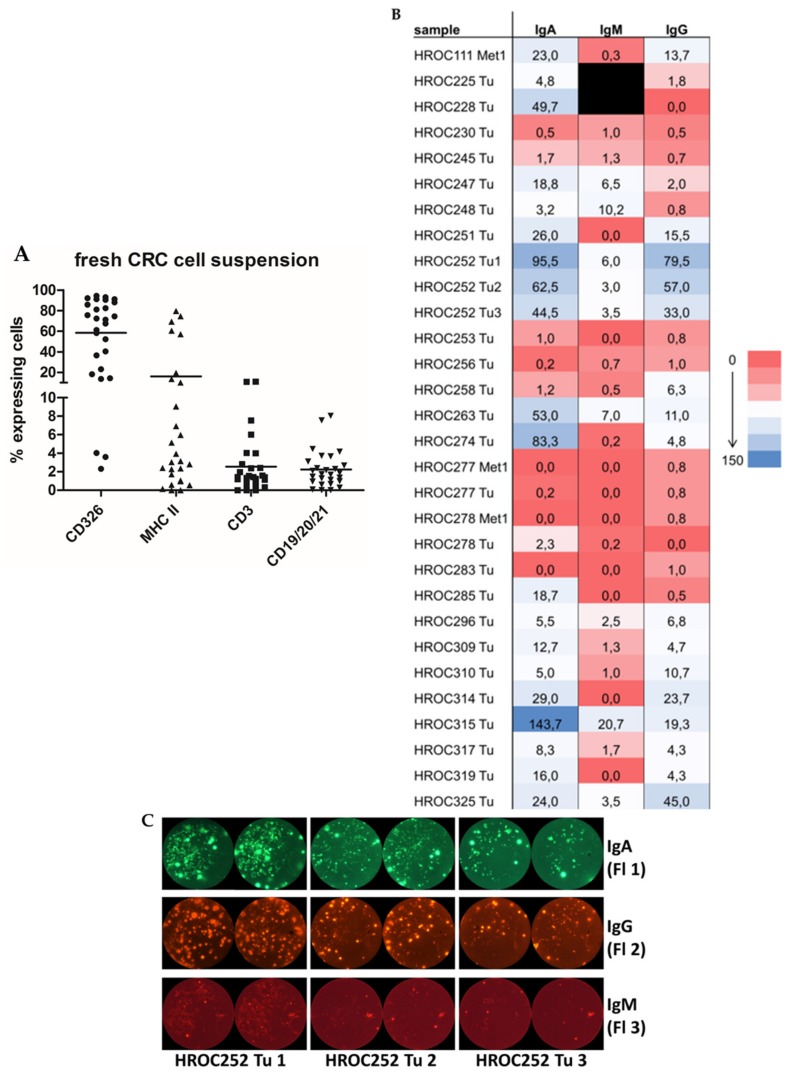
Cellular tumor composition and immunoglobulin secretion. (**A**) Composition of tumor cell suspensions. The cellular composition of the in situ analyzed colorectal carcinoma (CRC) specimen (*n* = 27/30) was assessed by flow cytometry using the following markers: CD326 (circle, epithelial (tumor) cells), major histocompatibility complex (MHC) class II (upward triangle, antigen-presenting cells), CD3 (square, T cells), and a lineage panel consisting of CD19, CD20, and CD21 (downward triangle, B cells). (**B**) Immunoglobulin (Ig) secretion of tumor cell suspensions. Numbers of IgA, -G, and -M secreting cells for all *n* = 30 cell suspensions are depicted in the heat map. Mean number of spots per sample and Ig type were plotted in a heat map using the conditional formatting function of Excel. The range goes from no secretors (dark red) to high numbers of secretors (dark blue). Met = metastasis; Tu = tumor; A = adenoma. Samples 225 and 228 could not be analyzed for IgM, thus the respective boxes are marked black. (**C**) Comparison of Ig secretion of three synchronous tumors. FluoroSpots representing IgA (green), -G (orange), and -M (red) secretors for tumors 1, 2, and 3 of patient HROC252 are depicted. The size of the original well of the FluoroSPot plate is 6.2 mm.

**Figure 2 cancers-11-00776-f002:**
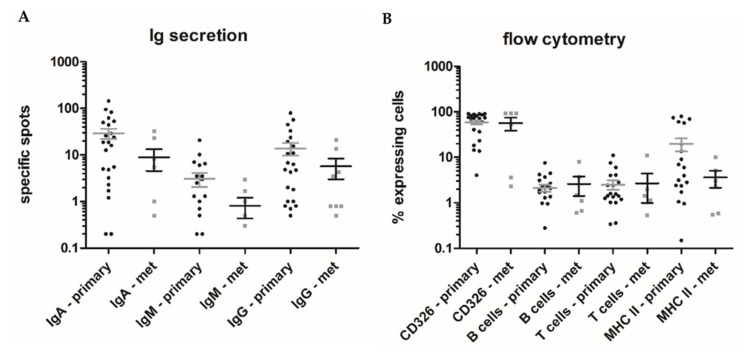
Comparison of primary and metastatic CRC samples. (**A**) Comparison of Ig secreting cells from primary CRC tumors and CRC-metastases. Numbers of IgA, -M, and -G secretors as assessed by FluoroSpot analyses are depicted for CRC tumors (black square) and CRC metastases (grey circle). (**B**) Comparison of tumor cell suspension composition from primary CRC tumors and CRC-metastases. Expression levels of CD326, B cells (CD19/20/21), T cells (CD3), and antigen-presenters (MHC II) as assessed by flow cytometry are depicted for CRC tumors (black square) and CRC metastases (grey circle).

**Figure 3 cancers-11-00776-f003:**
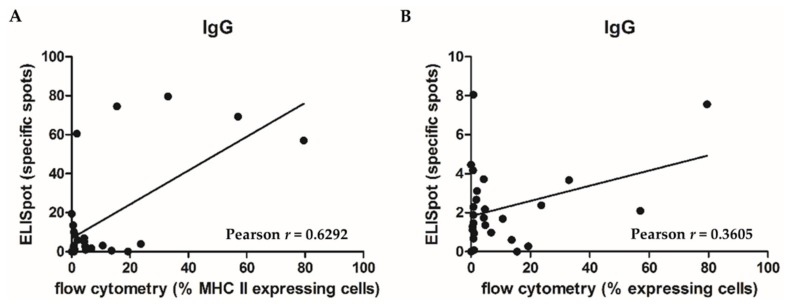
Correlation of Ig secretion with cellular tumor composition. (**A**) Correlation of IgG secretion and MHC II expression of CRC samples. The plot depicts the correlation between IgG secretion (on the *Y*-axis, assessed by FluoroSpot analyses) and MHC II expression (on the *X*-axis, assessed by flow cytometry) of CRC tumor cells. (**B**) Correlation of IgG secretion and B cell levels of CRC samples. The plot depicts the correlation between IgG secretion (on the *Y*-axis, assessed by FluoroSpot analyses) and B cell quantity (on the *X*-axis, assessed by flow cytometry using the lineage panel CD19/20/21) of CRC tumor cell suspensions.

**Table 1 cancers-11-00776-t001:** Patient Information.

Patient	Sex	Age	Tumor Site	TNM Stage	Tumor Grade	UICC Stage	Follow Up	
							Outcome	Years
HROC225	F	76	Rectum	T4 N2 M1 L0 V0	G3	UICC IV	dead	4.0
HROC228	M	54	Sigmoid	T3 N0 M0 L0 V0	G2	UICC IIa	tumor free	2.9
HROC245	M	73	Rectum	T3 N0 M0 L0 V0	G3	UICC IIa	tumor free	4.2
HROC247	M	74	right flexure	T3 N0 M0 L0 V0	G2	UICC IIa	dead	4.5
HROC248	M	65	Coecum	T4 N2 M1 L0 V0	G2	UICC IV	dead	2.3
HROC251	M	85	Ascending	T3 N0 M0 L0 V0	G1	UICC IIa	dead	2.2
HROC252	M	45	Descending	T4 N0 M0 L0 V1	G3	UICC IIb	tumor free	3.6
			Sigmoid	T4 N0 M0 L0 V1	G3	UICC IIb		
			Rectum	T4 N0 M0 L0 V1	G3	UICC IIb		
HROC256	M	70	Descending	T3 N0 M0 L0 V0	G2	UICC IIa	tumor free	4.3
HROC258	F	77	Recto-sigmoid	T3 N0 M0 L0 V0	G2	UICC IIa	dead	2.6
HROC274	M	64	Sigmoid	T3 N1 M1 L0 V1	G2	UICC IV	progression free	4.0
HROC277	M	77	Coecum	T4 N0 M1 L0 V1	G2	UICC IV	dead	3.3
			Liver metastasis					
HROC278	F	76	Ascending	T4 N2 M1 L1 V1	G3	UICC IV	dead	2.1
			Located peritoneal metastasis					
HROC283	F	48	Sigmoid	T3 N1 M0 L1 V0	G2	UICC IIIa	tumor free	3.4
HROC285	F	30	Descending	T4 N2 M1 L1 V0	G2	UICC IV	tumor free	3.7
HROC296	F	92	Ascending	T3 N0 M0 L0 V0	G2	UICC IIa	tumor free	3.6
HROC309	M	86	Descending	T3 N0 M0 L0 V1	G2	UICC IIa	dead	1.2
HROC310	M	76	Ascending	T3 N0 M0 L0 V0	G2	UICC IIa	tumor free	3.4
HROC314	F	76	Sigmoid	T4 N2 M1 L0 V0	G2	UICC IV	dead	2.9
HROC315	F	42	Left flexure	T3 N2 M0 L0 V1	G3	UICC IIIb	tumor free	2.6
HROC319	M	67	Coecum	T4 N2 M0 L1 V0	G3	UICC IIIb	tumor free	2.5
HROC325	F	80	Sigmoid	T3 N0 M0 L0 V0	G2	UICC IIa	tumor free	3.1
HROC230	M	74	Liver metastasis colon cancer			UICC IV	dead	6.2
HROC317	M	72	Liver metastasis colon cancer			UICC IV	dead	6.5
HROC253	M	61	Liver metastasis rectal cancer			UICC IV	dead	4.6
HROC111	M	78	Brain metastasis colon cancer			UICC IV	dead	3.1

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
