# Peer review of "Human Colorectal Carcinoma Infiltrating B Lymphocytes Are Active Secretors of the Immunoglobulin Isotypes A, G, and M"

_cancers, 2019, doi:10.3390/cancers11060776_

Reviewer 1 Report

The manuscript by Mullis et al showed that tumor infiltrating B lymphocytes secreting Ig’s has some significance.  However, the data do not seem to support conclusion that the authors have suggested particularly that these Ig’s are anti-tumor in nature.  No data were presented to show that these IgG binding to any tumor antigens.  In fact, the data albeit quite convincingly showed that there are tumor infiltrating B cells (TiB’s) in CRC’s but the presented data are insufficiently showing the function of these antibodies. As far as the significance of TiB’s expressing MHCII production of IgG, it is expected that most of the cells producing IgG or even IgM as APC’s; thus, one would expect them to express MHCII. 

It is recommended that prior to acceptance this paper, the authors demonstrate some function of these antibodies.  One suggestion they could do is to use these antibodies and see if they can pull down CD326+ cells.  

Minor points:  1)  It is very unusual to have the address of each author listed even though they are from the same institution (for example, affiliation #1).  2) The relationship between EBV and TiB's production of tumor specific IgG’s and current study is not clearly explained.  The readers should not have to pull out the old reference to get the authors’ points. 3) If the specimen are not Met = metastasis; Tu = tumor; A = adenoma, what are they?  Are they normal tissues?  Hyperplastic or Dysplastic cells?  4) It is not cleared how the specimen were analyzed for the "expression of Ig’s".  Is it gene expression or production of the IgG proteins?   I.e at first read, it looks like analysis was done by “gene expression” using mRNA blotted to a membrane but it looks like the membrane has proteins binding to IgA, IgG, or IgM.  Are the cells that were added to the membrane treated? Were the cells treated with Brefeldin A to block ER to Golgi transport.  Then, although cell suspension was used, the proteins bind to the membrane were in fact secreted proteins?  It is not clear  5)   What parameter(s) is/are used to determine whether it the Ig secretion is high, medium or low?  6) ELIspot is the same as Flurospot?

Major issues:    1) What are the epitopes of these antibodies?  Are these antibodies biding to the same antigens suggesting class switching?  2) Could these B cells produce in response to chemotherapy?  Are there controls for normal specimens or patients who have not been treate

Author Response

Response to Reviewer 1: 

It is recommended that prior to acceptance this paper, the authors demonstrate some function of these antibodies. One suggestion they could do is to use these antibodies and see if they can pull down CD326+ cells. 

We totally agree with the reviewer that this is one of the most interesting points in the context of TiBc properties. In our previous study [Maletzki et al., 2012 PLoSOne] on the functionality of TiBc-produced Igs, we delivered exactly this kind of data – a significant proportion of IgGs isolated from several TiBc clones recognized CRC cells. We did, however, not perform pull-down experiments but FACS-staining, IF-stainings and cell ELISA. In addition, we modified the introduction in order to make this clear to readers.

Minor points:

1. It is very unusual to have the address of each author listed even though they are from the same institution (for example, affiliation #1).

Affiliations were adapted accordingly.

2. The relationship between EBV and TiB's production of tumor specific IgG’s and current study is not clearly explained. The readers should not have to pull out the old reference to get the authors’ points.

Similar to the very first point, we adapted the wording and are confident that readers can easily get this point without having to read the previous work.

3. If the specimen are not Met = metastasis; Tu = tumor; A = adenoma, what are they? Are they normal tissues? Hyperplastic or Dysplastic cells?

We thank the reviewer for this attentive and helpful comment. This has been addressed and all specimens are now labeled with either Met, Tu or A.

4. It is not cleared how the specimen were analyzed for the "expression of Ig’s". Is it gene expression or production of the IgG proteins?  I.e at first read, it looks like analysis was done by “gene expression” using mRNA blotted to a membrane but it looks like the membrane has proteins binding to IgA, IgG, or IgM. Are the cells that were added to the membrane treated? Were the cells treated with Brefeldin A to block ER to Golgi transport. Then, although cell suspension was used, the proteins bind to the membrane were in fact secreted proteins? It is not clear

Again, we are grateful for this question. The technique of FluoroSpot (an enhanced version of ELISpot) we used in the present study has been explained in more detail in the improved version of the manuscript. However, technical details of the cell suspension preparation were not changed – the suspension was generated from fresh tumor specimen solely by mechanical dissociation and the medium used for the analysis contained no Brefeldin A or similar. Thus, the results obtained reflect to the best of our knowledge the intrinsic, unaltered properties of TiBc.

5. What parameter(s) is/are used to determine whether it the Ig secretion is high, medium or low?

Again, thanks for this helpful comment. The analysis procedure in Materials and Methods has been entirely rephrased in order to make this absolutely clear. Additionally, a short description has been added to the legend of Figure 1B.

6. ELIspot is the same as Flurospot?

This has been addressed in the answer to point 4.

Major issues: 

1.  What are the epitopes of these antibodies? Are these antibodies biding to the same antigens suggesting class switching?

Identifying epitopes recognized by TiBc-derived Igs, especially IgGs, is of utmost importance for unravelling the complex interaction network of the tumor microenvironment. There are only scarce reports on this topic and research activities are ongoing also in our laboratory to identify the antigens recognized by several of our TiBc clones-derived IgGs. However, this was clearly not the focus of the present work and we thus respectfully would like to omit this point.

2. Could these B cells produce in response to chemotherapy? Are there controls for normal specimens or patients who have not been treated.

Only a few patients enrolled in this study have been chemotherapeutically pre-treated. Thus, we can safely conclude that the production of Igs is not at all a consequence of chemotherapy.

Normal specimens have so far not been analyzed – partly due to the problem of collecting such tissues. However, we added a reference [Brandtzaeg et al., 1999] describing the proportion of B cells producing Igs of the subclasses A, M and G, resident in the normal gut (i.e. 90% IgA, 6% IgM and 4% IgG). We additionally discussed the differences between the normal distribution and our results in more detail and are confident that this point improved the scientific content of the manuscript substantially.

Reviewer 2 Report

The manuscript by Mullins et al. describes the phenotype and functions of tumor infiltrating B cells (TiBc) in colorectal carcinomas (CRC). Though this work highlights some of the regulatory features of TiBc in CRC, there are few concerns:

1.     The B cell data presented in line 82-84 is confusing. I wonder what methodology and markers were used to specify that the B cells defined are TiBc? I would suggest the authors show FACS plots of analysis to define the specificity of B cells.

2.     The authors show the secretion of IgA, IgG, and IgM by TiBc and described as high, intermediate and low levels. I wonder what was the basis of establishing the fact that these were specifically from TiBc? It would be nice to show the expression levels of these Ig in the serum of the CRC patients in order to have a better understanding.

3.     A good control group is missing in the data that the authors have presented. This lack of data makes the authors hypothesis very weak. The authors themselves have stated that this is one of the limitations of their manuscript. But seeing the strength of the data shown in this study, I believe that the aim of this work is yet not very clear.

Author Response

Response to Reviewer 2:

1. The B cell data presented in line 82-84 is confusing. I wonder what methodology and markers were used to specify that the B cells defined are TiBc? I would suggest the authors show FACS plots of analysis to define the specificity of B cells.

We thank the reviewer for this remark and added the information on the surface markers used to identify B cells out of the tumor cell suspension (i.e. CD19, CD20 and CD21 as lineage panel for B cells). In addition, we added two sentences to the discussion part describing the peri- or intratumoral localization of the cells from the tumors we analyzed. Due to the overall number of Figures already included into the short communication manuscript, we did not add FACS plots and the gating strategy. If finally requested, such data can of course be added.

2. The authors show the secretion of IgA, IgG, and IgM by TiBc and described as high, intermediate and low levels. I wonder what was the basis of establishing the fact that these were specifically from TiBc? It would be nice to show the expression levels of these Ig in the serum of the CRC patients in order to have a better understanding.

Here, we are a little unsure if we correctly understood the question of Reviewer 2. Thus, we try to address the different implications:

The analysis procedure in Materials and Methods has been entirely rephrased in order to make clear how the discrimination between low, intermediate and high was done. Additionally, a short description has been added to the legend of Figure 1B. Moreover, we clarified that the data presented in Figure 2A are numbers of Ig-secreting cells present in the tumor cell suspension. We would like to apologize for the use of the clearly wrong term “secretion levels” in the original version of the manuscript which might have caused this confusion.

How we ensured that the B cells analyzed can be safely regarded as TiBc has been answered in the previous point.

Finally, the reviewer suggests to show the levels of IgG, A and M from the serum of the patients. This is an interesting point but after careful consideration we would like to decline this request. The relevant control would be the situation in the normal gut and not in the peripheral blood.

Moreover, such a comparative analysis would be out of the focus of the current manuscript. But we will consider this point for future analysis - sera of the respective patients have been collected at the time of operation and could be analyzed concerning Ig subclass levels.

3. A good control group is missing in the data that the authors have presented.

Normal specimens have so far not been analyzed – partly due to the problem of collecting such tissues. However, we added a reference [Brandtzaeg et al., 1999] describing the proportion of B cells producing Igs of the subclasses A, M and G, resident in the normal gut (i.e. 90% IgA, 6% IgM and 4% IgG). We additionally discussed the differences between the normal distribution and our results in more detail and are confident that this point improved the scientific content of the manuscript substantially.

Reviewer 3 Report

-is there evidence of infiltrating b cells in tumors in mouse models? this type of literature could help improve the scientific soundness between the intro and the discussion

I am not sure what the authors mean by "immunome" the term, although not widely used, refers to deep immune profiling using more than 10 parameters. this terms should be changed e.g cell pupations. 

results 2.2 2.3 and 2.4 should be combined into one as the text with flow much better rather than have separate sections of the same related results, I would also consider giving a more appropriate title to the combined result 

for fig 1B it is not clear at all how the secretion was measured. is this a fold change? if not, I don't think a heat map is the way to express the amount of secretion. Is it physiologically relevant to compare IgA secretion to IgM secretion? instead of IgA to IgA etc.. a good way to confirm these up/downregulation of the Ig's would be to run a qPCR on the samples for IgA, IgM, IgG and other B cell markers (CD19, 21) to confirm activation. samples HCRO225 and 228 is there a reason why IgM is in black, does this mean lack of readout? because there are several other readouts that are at 0. 

Figure 1C. It seems like all 3 markers are colocalizing. was there a positive/negative control ran with the samples? is it possible that what we are seeing is background staining? please include positive and negative controls.

Figure 2A/B, if the axes in the figures were to be shown on a log scale, this would make it easier to appreciate the lower values. please include error bars for all your plots either SEM or SD and indicate which one you are using. 

figure 3A/B please show the coefficient of determination of the correlations along with the fitted trend line 

It has been repeated in the text that the study "fails to reach statistical significance" however there is no mention of what statistics were used, nor what type of groups were analyzed. 

Author Response

Response to Reviewer 3:

1. is there evidence of infiltrating b cells in tumors in mouse models? this type of literature could help improve the scientific soundness between the intro and the discussion

We want to thank the reviewer for this remark. There are reports on a tumor-promoting role but also reports on a clear role of TiBc in the fight against cancer from different mouse models. A recent report was the first to shed light into this controversial by showing that this might be simply due to intrinsic differences in the models used [Spear et al., 2019]. We included the major findings into the discussion.

2. I am not sure what the authors mean by "immunome" the term, although not widely used, refers to deep immune profiling using more than 10 parameters. this terms should be changed e.g cell pupations. 

This has been done as requested.

3. results 2.2 2.3 and 2.4 should be combined into one as the text with flow much better rather than have separate sections of the same related results, I would also consider giving a more appropriate title to the combined result 

This has also been done.

4. for fig 1B it is not clear at all how the secretion was measured. is this a fold change? if not, I don't think a heat map is the way to express the amount of secretion. Is it physiologically relevant to compare IgA secretion to IgM secretion? instead of IgA to IgA etc. a good way to confirm these up/downregulation of the Ig's would be to run a qPCR on the samples for IgA, IgM, IgG and other B cell markers (CD19, 21) to confirm activation. samples HCRO225 and 228 is there a reason why IgM is in black, does this mean lack of readout? because there are several other readouts that are at 0. 

This remark from the reviewer is very helpful. First, the technique of FluoroSpot (an enhanced version of ELISpot) we used in the present study has been explained in more detail in the improved version of the manuscript. The data presented are not fold change - in order to make this easier to understand, the analysis procedure in Materials and Methods has been entirely rephrased. Additionally, a short description has been added to the legend of Figure 1B.

Second, there is a little misunderstanding in the nature of our quantitative analysis. We would like to apologize for the use of the clearly wrong term “secretion levels” in the original version of the manuscript which might have caused this confusion. We did not measure “amount of secretion” but “amount of secreting cells”. This has been changed to the correct terminology all over the manuscript. “Amount of secreting cells” can be scientifically correctly displayed in a heat map – similar data on T cells have for example been repeatedly published [for example Schwitalle et al., 2008 in Gastroenterology].

Concerning the “black” IgM results of samples HROC225 and HROC228, we thank the reviewer for his attentive remark. Indeed, due to technical problems there were no reliable data and were consequently omitted. We added the respective information into the figure legend of Figure 1B.

5. Figure 1C. It seems like all 3 markers are colocalizing. was there a positive/negative control ran with the samples? is it possible that what we are seeing is background staining? please include positive and negative controls.

Again, this is a very attentive remark. We can safely exclude co-localization since the different Igs were measured in different wells. Positive and negative controls have been performed using either TiBc clones producing the respective Ig subtype (positive controls) or without the addition of cells to the medium used in the FluoroSpot analysis (negative controls). These controls are necessary to train the readout software. Adding pictures of the respective control wells into the Figure 1 C will in our opinion, however, potentially cause more confusion for most readers then helping to clarify. Thus, we decided to keep Figure 1 C unaltered – but are prepared to deliver a modified version if finally asked for.

6. Figure 2A/B, if the axes in the figures were to be shown on a log scale, this would make it easier to appreciate the lower values. please include error bars for all your plots either SEM or SD and indicate which one you are using. 

This has been done according to the reviewer’s request.

7. figure 3A/B please show the coefficient of determination of the correlations along with the fitted trend line 

We again added the information asked for.

8. It has been repeated in the text that the study "fails to reach statistical significance" however there is no mention of what statistics were used, nor what type of groups were analyzed. 

We want to thank the reviewer for this helpful hint and added the information as requested.

Round  2

Reviewer 2 Report

I thank the authors for answering all my queries

Reviewer 3 Report

I would like to thank the authors for taking into consideration my remarks and explaining things over. the corrections are satisfactory.